# The Unique Pharmacometrics of Small Molecule Therapeutic Drug Tracer Imaging for Clinical Oncology

**DOI:** 10.3390/cancers12092712

**Published:** 2020-09-22

**Authors:** Mark P. S. Dunphy, Nagavarakishore Pillarsetty

**Affiliations:** 1Department of Radiology, Memorial Sloan Kettering Cancer Center, New York, NY 10065, USA; pillarsn@mskcc.org; 2Department of Radiology, Weill Cornell Medical College, New York, NY 10065, USA

**Keywords:** theranostics, imaging biomarkers, diagnostic biomarkers, predictive biomarkers, small molecule drugs, pharmacokinetics

## Abstract

**Simple Summary:**

New clinical radiology scans using trace amounts of therapeutic cancer drugs labeled with radioisotope injected into patients can provide oncologists with fundamentally unique insights about drug delivery to tumors. This new application of radiology aims to improve how cancer drugs are used, towards improving patient outcomes. The article reviews published clinical research in this important new field.

**Abstract:**

Translational development of radiolabeled analogues or isotopologues of small molecule therapeutic drugs as clinical imaging biomarkers for optimizing patient outcomes in targeted cancer therapy aims to address an urgent and recurring clinical need in therapeutic cancer drug development: drug- and target-specific biomarker assays that can optimize patient selection, dosing strategy, and response assessment. Imaging the in vivo tumor pharmacokinetics and biomolecular pharmacodynamics of small molecule cancer drugs offers patient- and tumor-specific data which are not available from other pharmacometric modalities. This review article examines clinical research with a growing pharmacopoeia of investigational small molecule cancer drug tracers.

## 1. Introduction

Imaging in vivo pharmacokinetics and drug biodistribution, using isotopologues or radiolabeled analogues of chemically synthesized small molecule (<900 Dalton molecular weight) pharmacotherapeutics, administered in microdose (<100 micrograms, with no pharmacologic effects), radiotracer (delivered patient radiation absorbed doses comparable to those from standard radiologic scans) amounts, referred herein simply as drug tracers, is much advocated and increasingly realized in optimizing translational and early clinical phase drug development, with great potential for personalizing cancer care [1,2,3,4,5,6] (Figure 1).

Developing new cancer drugs is a slow, expensive process that usually fails [11,12,13]. The need for new target- and drug-specific pharmacometrics is recurrent and fundamental to the clinical development of new cancer pharmacotherapeutics [14,15]. A growing pharmacopoeia of investigational drug tracers have been clinically tested, with various degrees of qualification [1] as imaging biomarkers (Table 1).

Extensive published research indicates the potentially revolutionary impact of drug tracer imaging as a fundamentally new kind of pharmacometric modality, offering unique, high-impact, patient-, tumor-, and drug-specific in vivo data that can both optimize patient outcomes and economize drug development [56]. National organizations are working to facilitate validation of these and other radiologic biomarkers for the future of patient care [1,3,57,58,59,60,61,62].

The vast majority of published drug tracer research has focused on positron emission tomography (PET) imaging, which can detect picomolar amounts of radiolabeled drug in tissues in vivo [63,64]. In this article, we focus on clinical research with drug tracers developed as drug-specific companion diagnostics for cancer therapy, e.g., ^18^F-5-fluorouracil, a drug tracer studied for the chemotherapeutic small molecule drug 5-fluorouracil [19]. We discuss the potential of drug tracer imaging to optimize patient selection, dosing strategy, and pharmacometrics in targeted cancer therapy with small molecule drugs in ways that conventional biomarker modalities cannot [14,15,65].

## 2. What Does Tumor Signal on a Drug Tracer Image Signify?

Medical imaging with drug tracers and other radiotracers typically seeks to detect high tumor signal, relative to background or other reference tissues, commonly termed avidity (Figure 2). Avidity, on imaging, is a net result of biologic factors affecting drug influx, efflux, and retention in tumors [65,66] and of tracer in vivo radiopharmacology (including effective half-life) and imaging technology [67] that affect the detectability of tumor avidity on imagery. Tumor avidity for drug tracers often depends upon strong drug-target binding, i.e., irreversible or slow drug-target dissociation (k_OFF_). Types of drugs with rapidly reversible drug–receptor binding might not be able to produce a simple tumor avidity “hotspot” signal useful in routine radiology practice as diagnostic or predictive biomarkers, instead requiring more complex image acquisitions and data analyses [63].

Tumor avidity can be a diagnostic imaging biomarker of the molecular target to which the drug tracer binds in vivo. Tumor avidity for the isotopologue I-124 PU-H71 is a well-credentialed imaging biomarker of tumor epichaperome formations [49,68]. Epichaperomes are hyperconnected, integrated functional networks of chaperome units, induced by cellular stress as associated with malignancy [49,68,69,70]. For drugs with multiple molecular targets, tumor avidity might have in vivo specificity as a diagnostic biomarker of one drug target if a particular cancer assayed predominantly expresses one target [9,71].

Absence of tumor avidity (non-avidity) can indicate that the tumor does not express the molecular target or that other in vivo factors prevent the drug tracer from reaching and/or binding to the target [72]. For example, in vivo tumor avidity for F-18 paclitaxel varies inversely with tumor expression of the multidrug resistance (MDR) drug efflux pump; higher tumor F-18 paclitaxel avidity is associated with higher in vivo tumor drug sensitivity [30].

## 3. Drug Tracer Imaging to Guide Patient Selection

Establishing tumoral target expression is key for rational patient selection for a targeted therapeutic drug. Biopsy-based histologic assay is the conventional gold standard diagnostic biomarker of target expression, but small clinical studies suggest that drug tracer imaging can be a superior predictive biomarker of drug efficacy [30,72].

Our group developed a radiolabeled analog of the therapeutic tyrosine kinase inhibitor dasatinib [9] for PET imaging. Tumor resistance to dasatinib can involve altered tumor pharmacokinetics [73,74,75,76,77,78,79]. Resistant tumors may actively remove dasatinib (e.g., by efflux pumps) [73,74,75] or have mutations in kinase targets that prevent dasatinib binding [73,74,75,80].

Imaging with radiolabeled octreotide fails to detect somatostatin receptor (SSTR) expression in some tumors, but for tumors overexpressing SSTR, on histology, octreotide tracer avidity appears to stratify tumor responses to octreotide therapy [81,82].

The tyrosine kinase inhibitor erlotinib binds to epidermal growth factor receptor (EGFR) exon 19 deletions or exon 21 (L858R) substitution mutations, called sensitizing mutations, affecting the erlotinib binding site. Bahce et al. demonstrated that tumors with sensitizing mutations concentrate [C-11]-erlotinib better than tumors lacking those key mutations—despite equivalent EGFR protein expression—with non-avid tumors progressing more quickly [36].

Tumor avidity for F-18 radiolabeled paclitaxel analogue (FPAC) correlated with in vitro paclitaxel sensitivity in animal models [83] and appeared to correlate with in vivo tumor sensitivity in a small group of human subjects [30,84].

For novel therapeutic drugs that target based upon kinetic selectivity, a simple tissue-based (e.g., immunohistochemistry) target assay might not exist [85]. For example, PU-H71 kinetically selects for epichaperome-integrated HSP90; no epichaperome IHC assay exists and histologic HSP90 expression is not predictive. Tumor avidity for PU-H71 tracer predicts pharmacodynamics and efficacy for PU-H71 [7,8,49,68,69,86] and other small molecule drugs targeting the HSP90 ATPase and interfering with epichaperome formation [8,49].

## 4. Tumor Pharmacokinetics for Image-Guided Dosing Design

Drug tracer imaging can assay tumor pharmacokinetics, including (1) tumor drug concentrations as a function of time post-drug administration and (2) tumor occupancy by therapeutic drug doses [87]. Tumor pharmacokinetic data from patient imaging can be mathematically modeled for predicting the therapeutic dosage and schedule needed to achieve desired in vivo tumor drug concentration and tumor saturation levels [8,49] (Figure 3). Imaging biodistribution can help to establish correlations with observed organ toxicity that might enable dosing strategies optimized for the therapeutic index [65].

Pioneering clinical research in imaging in vivo tumor pharmacokinetics studied the cytotoxic drug 5-fluorouracil both by PET [88,89,90] and by in vivo nuclear magnetic resonance spectroscopy (NMRS) [18,20,21,91,92,93] and explored the microdose versus therapy dose biodistribution and pharmacokinetics of radiolabeled N-[2-(dimethylamino)ethyl]acridine-4-carboxamide (DACA) [94]. Saleem et al. demonstrated in cancer patients that the prodrug temozolomide undergoes in vivo metabolic conversion to its active form, by studying the PET isotopologue [^11^C]temozolomide and its in vivo radiometabolites [17]. Saleem and Price [89], analyzing accumulated PET pharmacokinetic data used in previously published studies [16,90,94], evaluated the influence of tumor perfusion on tumor pharmacokinetics.

We and others have explored clinical PET-based approaches to quantifying in vivo tumor drug molar concentrations [8,37,49,95]. Ideally, for tracing pharmacokinetics, drug tracer and (non-labeled) therapeutic drugs are isotopologues, as drug and tracer should then have identical physicochemical characteristics (Figure 1). Microdose (tracer) versus therapy dose in vivo pharmacokinetics, metabolism, and biodistribution can differ significantly, e.g., including potential tumor saturation affecting the accuracy of microdose predictions of tumor molar concentrations achieved by therapy doses. In our trials [7,8,49], we have explored therapy dose (mixing therapeutic drug and tracer together, NCT01393509) and microdose drug tracer PU-H71 imaging (NCT01269593). We validated the quantitative accuracy of non-invasive PET-based measurements of tumor PU-H71 molar concentrations, after administering therapeutic (non-radioactive) PU-H71 dose mixed with drug tracer, using liquid chromatography–tandem mass spectrometry (LC-MS/MS) of PU-H71 concentrations in biopsy specimens (NCT01393509 [8]).

Tumor avidity quantified for derivation of molar concentration must account for any tumor-residualizing radiometabolites. A drug (tracer) might have one or more active metabolites; drug tracer metabolites, produced in vivo, might be radiolabeled or unlabeled. In vivo radiometabolites can be detected by plasma sampling or, for some carbon-11 (C-11) drug tracers, in assayed exhalations [17]. I-124 PU-H71 produces three radiometabolites in vivo but these all spontaneously efflux from the tumor, simplifying tumor analysis. PU-H71 therapeutic drug and its isotopologue are both given intravenously, also simplifying our modeling of drug molar concentrations. Small molecule cancer drugs are often orally administered, with less than 100% bioavailability (compared to the 100% bioavailability of an IV-administered drug). Imaging after oral administration of a drug tracer potentially enables correlation of gastrointestinal drug transit, plasma concentrations, and tumor concentrations.

Conventional selection of a dosing strategy in early-phase therapy trials is based on preclinical PK models to predict sufficient tumor exposure to drug [65]. Our group established a preliminary dose–response relationship for PU-H71 in tumor in vitro cultures, identifying a threshold drug concentration time integral necessary to collapse tumor epichaperomes that correlated with target saturation [49,86]. We next validated the preclinical correlation between tumor pharmacokinetics and tumor pharmacodynamics in vivo, using PET and tissue assays in animal tumor models [86], and now we have explored these PK/PD correlations in cancer patients [7,8,49]. Our clinical PET imaging of tumor kinetics confirmed that tumor C_max_ is key, as expected for dose-dependent responses, but our clinical data indicate that in vivo tumor concentrations required for anti-tumor efficacy are an order of magnitude higher than predicted by our prior in vitro data [8]. The PET has helped us to select a phase 2 (NCT03166085) PU-H71 therapeutic dosing strategy that we expect will saturate epichaperome-positive tumors efficaciously [8,49].

Drug tracer imaging enables a non-invasive clinical approach to quantifying in vivo tumor occupancy by therapeutic drug [37,96]. Tumor occupancy or complete saturation by drug uptake are indicated by decreased tumor uptake of drug tracer in the presence of a co-administered therapeutic drug dose, compared to tumor uptake of microdose tracer alone [37,87] (Figure 3). Tumor saturation does not necessarily indicate that all tumor molecular targets are bound by the drug; it can also indicate saturation or alteration of tumor pharmacokinetics [17]. Imaging therapeutic drug inhibition of drug tracer uptake can be used to identify the maximum single therapeutic dose that a tumor can accumulate and retain. Drug tracer imaging can detect whether or not the maximum tolerated dose of a therapeutic drug, determined empirically in phase 1 therapy dose escalation trial, is able to saturate tumors in vivo [16]. Clinical C-11 erlotinib PET imaging demonstrated variable saturation of in vivo tumor EGFR targets in NSCLC patients receiving erlotinib therapeutic doses [37].

Drug tracer imaging potentially could be incorporated into dose escalation therapy trials to detect whether tumor saturation, defined by imaging, occurs prior to reaching the maximum tolerated dose (MTD) level, defined by patient toxicity. Hypothetically, once tumor saturation is achieved, higher therapeutic dose levels might not yield proportional, incremental increases in tumor drug concentrations or responses, despite increasing dose-dependent patient toxicity [73,97,98,99]. Pharmacodynamic studies often reveal a sigmoidal dose-response curve, with “diminishing returns” at higher dose levels [100,101]. If a non-MTD, tumor-saturating dose level, which we call a maximum tumor dose, is detected by imaging, researchers could then pursue identifying the maximum tolerated schedule for that saturating dosage, modeling which steady-state peaks and trough tumor molar concentrations would be obtained [8,49,68]. For example, a lower therapeutic dose of the small molecule tyrosine kinase inhibitor dasatinib, given more frequently, was equally efficacious against leukemia and less toxic to patients, as a less frequent, higher dose strategy [102]. Drug tracer imaging is the only available modality of biomarker assay that enable clinicians to explore this novel dose-learning approach, based upon in vivo tumor pharmacokinetics, in cancer patients.

Drug tracer imaging can aid in the design of combination therapy regimens, in studying drug interactions and pharmacodynamics. Drug tracer imaging revealed an unanticipated drug interaction in an investigational therapy regimen that combined the antiangiogenic therapeutic drug bevacizumab with the cytotoxic drug docetaxel: imaging demonstrated reduced tumor uptake of (C-11) docetaxel isotopologue for four days after bevacizumab administration; these imaging data helped to guide a redesign of this dosing strategy [103]. Saleem et al. [90] studied the effects of a pharmacological intervention (eniluracil modulation) on the in vivo kinetics of the cytotoxic agent 5-fluorouracil in tumors and healthy organs by 5-[^18^F]-FU drug tracer imaging.

In non-human primates, Kurdzeil et al. [30] used F-18 paclitaxel (FPAC) PET to detect the in vivo effects of tariquidar, an inhibitor of the P-glycoprotein (Pgp) efflux pump that is implicated as a moderator of multidrug resistance (MDR), on paclitaxel kinetics. Tariquidar did not alter FPAC plasma kinetics, but PET successfully detected a change in FPAC biodistribution consistent with Pgp inhibition.

## 5. How to Make a Drug Tracer

Synthesis of PET or single photon emission computed tomography (SPECT) (i.e., radiolabeled) versions of therapeutic small molecules is usually a more challenging process than, for example, synthesis of radiolabeled versions of therapeutic antibodies. Relatively simple techniques are well-established such that each can radiolabel a wide variety of antibodies; in contrast, synthesizing a radiolabeled version of a drug or tracer that is closely related, structurally, to a drug of interest requires radiochemistry expertise to design a synthetic process that will generate a specific precursor molecule that can be radiolabeled to yield the desired radiotracer. For a drug analogue, when installing a radioisotope, a key design criterion is that the position and type of radioisotope must not significantly compromise the binding affinity of the drug analog to the molecular target(s). Much effort is spent optimizing the synthetic scheme so that radiolabeling can be done at the latest possible synthetic step to minimize potential side products and isotope decay. In general, most PET tracers containing carbon-11, nitrogen-13, fluorine-18, and iodine-124 are synthesized in a two-step method involving radiolabeling of a suitable radioactive precursor followed by deprotection. This is followed by purification, isolation, and formulation in a biocompatible buffer. In considering the position of the radiolabel on the small molecule, the general rule of thumb is that it must be both chemically and metabolically stable. For example, alkyl iodides and benzylic (-CH_2_) fluorides are highly labile and therefore not common in PET drugs. The available supply and chemical form of the radioisotope dictate the type of chemistry amenable to installing the radiolabel for subsequent in vivo use. Carbon-11 is routinely available as CO_2_ or CH_4_ (50–200 GBq) and has to be converted to a suitable reactive form to be installed onto a small organic molecule. Fluorine-18 is commonly available as a non-reactive fluoride (F^-^) (50–200 GBq) in water and must be completely dried and activated before it is incorporated into an organic molecule using nucleophilic (S_N_2) displacement or S_N_Ar type reactions. Recent developments in metal-catalyzed aromatic substitution reactions have rapidly expanded the scope of the chemistry with [^18^F]-F^-^ and have facilitated the production of aromatic- and heterocyclic-fluorinated PET tracers that were previously considered inaccessible with [^18^F]-F^-^. Iodine-124 is available as iodide (I^-^) (0.5–5 GBq) and can be easily installed on a suitable aromatic backbone using direct electrophilic iodination of activated arenes or via a variety of iodo-demetallation reactions on inactivated arene backbones.

In developing drug tracers for human studies, a suitable radiosynthetic method and precursor must facilitate (1) easy, scalable, and reproducible radiochemical yields, (2) automation, (3) easy purification, and (4) formulation in a biocompatible solvent. The radiochemical precursor is a suitable compound whose chemical manipulation with radioactive elements yields the intermediate product that can be converted to the desired final radioactive product or is sometimes the final product itself. The precursor must be reactive enough to facilitate chemical/radiochemical manipulations but stable enough to be vialed and stored for future use (up to two years) without loss of radiochemical yields, isomerization, or chemical transformation that can lead to undesirable products. The precursor and its breakdown products during radiochemical reactions must be easily separable from the final radiochemical compound to ensure that the final product is not only radiochemically pure but also chemically pure. An additional consideration in the precursor design is whether the compound contains toxic elements (e.g., tin, lead, or mercury); a suitable analysis method must be developed to quantify concentrations in the final product. In developing a suitable radiosynthetic method, solvent optimization is essential for each step to address logistical challenges related to automation, purification, and/or formulation due to compatibility, miscibility, corrosiveness, high boiling points, etc. Minimizing the use of HPLC is also advantageous, as is ensuring that separation of the final product can be achieved through physical manipulation.

Once a reproducible synthesis of a candidate drug tracer has been developed, this candidate is subjected to a battery of in vitro and in vivo tests. In vitro testing must demonstrate high affinity binding/avidity of the drug tracer to its target and its ability to be blocked by a suitable competitor, including the drug itself. For example, in designing F-18 SKI-249380, a tracer of the small molecule therapeutic drug dasatinib, we placed a positron-emitting fluorine isotope (F-18) atom to substitute for a hydroxyl group [9,10,71]. Molecular modeling predicted that the substitution had a minimal effect upon drug binding to target kinases, which we confirmed by comparing in vitro pharmacodynamic profiles and potencies and in vivo imaging of competitive inhibition of tracer uptake by a co-administered dasatinib therapeutic dose.

Another important criterion is to ensure that the drug tracer has minimal non-specific binding in the cell or system. This is followed by in vivo testing, in suitable animal models, to demonstrate drug tracer uptake by the tumor or other target-containing tissue(s). This becomes extremely important for privileged organs such as the brain, cerebrospinal fluid (CSF), or tumor, where barriers (e.g., blood–brain barrier, interstitial pressure, low vascularity, etc.) can severely restrict penetration of the drug into the tissue of interest. If the animal imaging studies demonstrate feasibility of using the candidate molecule as a drug tracer, the blood half-life and metabolic profile of the compound is then assessed to demonstrate that the signal from the tissue of interest is arising due to specific binding of the drug tracer to its target rather than merely due to blood pool activity (e.g., high plasma protein binding) or a radioactive metabolic byproduct. For the promising candidate tracer, animal dosimetry studies then measure radioactive dose to organs (mouse); the results are then extrapolated to predict human absorbed and effective doses. To pursue clinical translation to first-in-human studies, further requirements typically include animal toxicology studies, GMP-quality radiochemistry synthesis, radiotracer synthesis validation runs, and institutional and regulatory agency (e.g., FDA) approval of a clinical research protocol embedded within an investigational new drug application package. Radiolabeled small molecule drugs are usually synthesized with a high specific activity, meaning that patients receiving the radiotracer injection, sufficient for imaging of drug biodistribution, often receive only a “microdose” of drug (<100 micrograms). As such, small molecule drug tracers for microdose imaging typically demonstrate no pharmacodynamic effect or pharmacologic toxicity.

## 6. Does Drug Tracer Imaging Offer Any Advantages over Conventional Tissue-Based PK/PD Biomarkers?

Drug tracer imaging offers potentially revolutionary, multifaceted patient-, tumor-, drug-, and target-specific biomarker data that tissue-based assays cannot, including tumor saturation, drug molar concentrations and kinetics, tumor heterogeneity, and biodistribution data.

Plasma pharmacokinetics, though essential data, often fail to predict tumor pharmacokinetics and response, for which tracer imaging of tumor pharmacokinetics might be a better predictive biomarker [6,8,49,90,98,99,104]. For PU-H71, we found that not only is the maximal tumor concentration achieved important for drug efficacy, but maintaining threshold tumor concentrations for a threshold duration of time was also key [86], whereas plasma pharmacokinetics did not differ significantly between responders and non-responders [8]. For drug administration to specific compartments (e.g., intrathecal, intraperitoneal, intratumoral), plasma pharmacokinetics do not provide a reliable surrogate of drug concentrations and distributions within those spaces and immediately surrounding tissues. The ability to correlate tumor response to tumor dose, not merely to the administered or plasma dose, advances the precision of dose–response analyses from patient- to lesion-level [88,105].

Tumor heterogeneity refers to significant intra- and inter-tumor variations in molecular target expression. Quantifying target expression by imaging of drug tracer uptake throughout a single tumor, as well as in all tumors in a given patient, might achieve stronger correlations with tumor response and insights into mixed tumor responses [72]. Needle biopsy collects biomarker data from only a small sliver of a tumor. Biofluid biomarkers or surrogates, such as circulating tumor cells, offer only a general correlate of systemic tumor burden and biomarker expression.

Drug tracer imaging enables four-dimensional whole-body assessment of drug tracer biodistribution across serial scan time-points. Biodistribution imaging data can reveal unexpected drug tracer accumulation in certain organs that might prompt serum or other assays to monitor for toxicity in that organ during clinical therapy trials, potentially improving patient safety and clarifying drug toxicity profile [106]. Biodistribution might identify compartments of poor drug penetration. PET isotopologues have been used to quantify the blood–brain barrier penetration of small molecule drugs in early-phase development [35,37,40], thereby predicting efficacy against brain metastases and encouraging further testing [34]. Clinical research with a temozolomide drug tracer, when temozolomide was an investigational therapeutic, demonstrated drug uptake in glioma brain tumors with scant drug retention by normal brain, predicting a favorable therapeutic index [17]. Similar clinical data were obtained with a lapatinib drug tracer in breast cancer patients with brain metastases [107,108].

## 7. Conclusions

Clinical PET assays of tumor targeting by small molecule drugs provide a fundamentally new mode of perception for cancer therapy investigators, enabling clinical testing of new paradigms in targeted drug development. As a pharmacometric modality, clinical PET imaging-based microdose studies, using a trace (<100 micrograms) amount of drug isotopologue or radioanalogue, offer patient-, drug-, and tumor-specific in vivo data that conventional plasma- or biopsy-based pharmacokinetic and biomarker assays cannot.

## Figures and Tables

**Figure 1 cancers-12-02712-f001:**
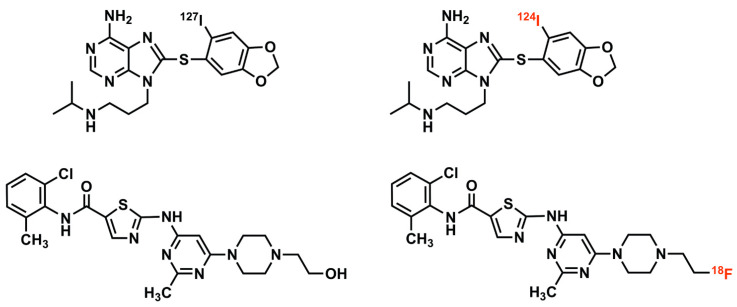
Drug tracer imaging of small molecule therapeutic compounds employs isotopologues or radiolabeled analogues. For example, I-124 PU-H71 (top right) is an isotopologue of the epichaperome inhibitor therapeutic compound, PU-H71 (top left), identical in molecular structure, differing only in the isotopic form of a constituent atom (stable iodine-127 versus the positron-emitting iodine-124) [7,8]. F-18 SKI-249380 (bottom right) is a radiolabeled analogue of the tyrosine kinase inhibitor therapeutic compound, dasatinib (bottom left). The positron-emitting fluorine-18 atom, in F-18 SKI-249380, replaces a hydroxyl group found in dasatinib; this substitution has minimal pharmacologic effect, enabling F-18 SKI-249380 to be an effective drug tracer of dasatinib [9,10].

**Figure 2 cancers-12-02712-f002:**
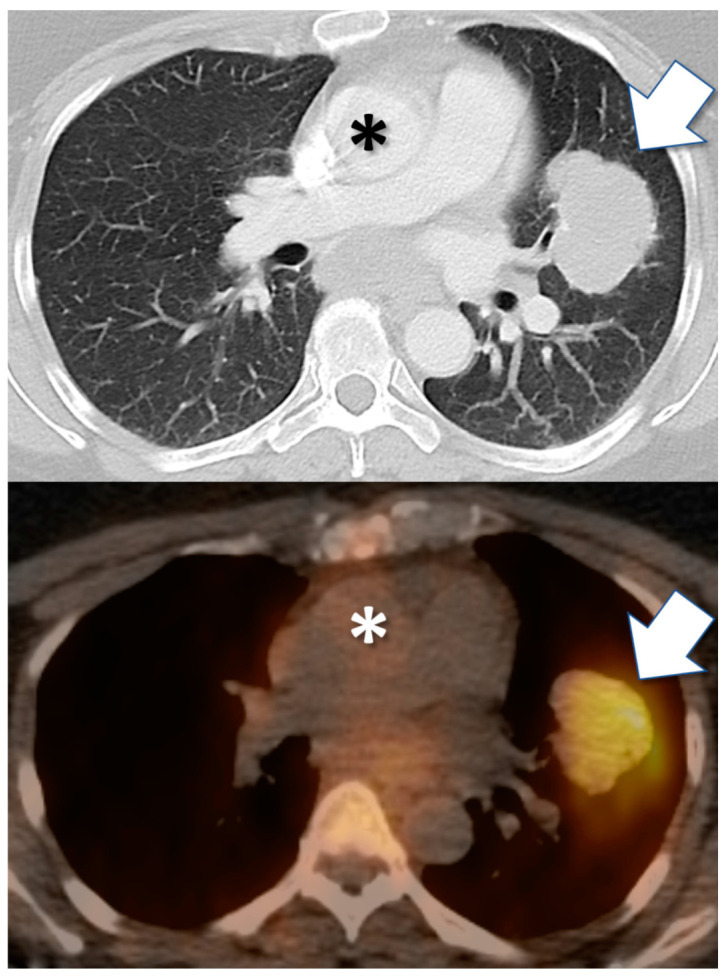
Drug tracer imaging detects in vivo tumor avidity for therapeutic small molecule compounds. Axial CT (top) and PET/CT fusion (bottom) images from 57-year-old female with clear cell carcinoma of Müllerian origin, including biopsied malignant pulmonary mass (arrow). PET/CT image obtained 24 h after injection with I-124 PU-H71 radiotracer, an isotopologue of the epichaperome inhibitor therapeutic compound PU-H71. PET demonstrates marked retention of drug by tumor, after drug has cleared from blood circulation (e.g., cardiac blood pool, *asterisk*).

**Figure 3 cancers-12-02712-f003:**
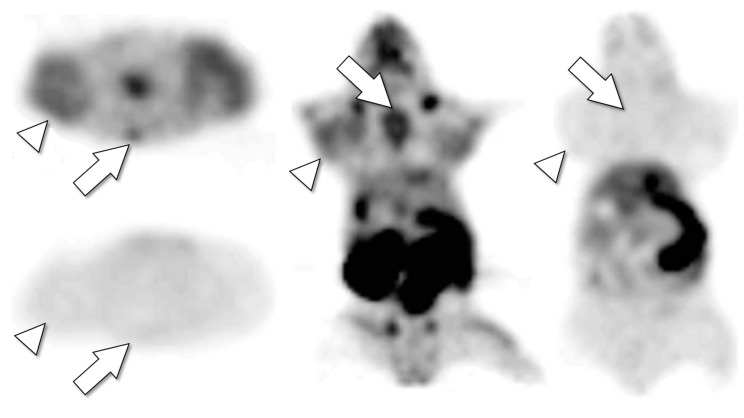
Drug tracer imaging can detect tumor saturation by therapeutic drug. Axial (left top, left bottom) and coronal (middle, right) PET images of single mouse bearing tumors after IV injection with F-18 SKI249380, radiolabeled analogue of therapeutic tyrosine kinase inhibitor dasatinib. Microdose (radiotracer only) images (left top, middle) detect avid tumors (*arrowheads*, H1975 lung cancer xenograft) and bone marrow (e.g., sternum, *arrows*). The next day (after F-18 isotope decay), the mouse was re-imaged after receiving F-18 SKI249380 radiotracer co-injected with therapeutic dose of dasatinib (left bottom, right). The therapeutic dasatinib dose competitively inhibited uptake of the radiotracer in tumors and bone marrow, indicating that the drug and tracer share the same in vivo tumor pharmacokinetic pathways and that these drug-specific tumor pharmacokinetic pathways (cellular kinase targets and/or pre-target uptake and transport mechanisms) were saturated by the single therapeutic dose. The tumor-saturating dose used in the animal model (30 mg/kg) was equivalent (by allometric extrapolation) to a 140 mg human dose, less than the maximum tolerated dose of some solid tumor dasatinib therapy trials [73].

**Table 1 cancers-12-02712-t001:** Selected examples of isotopologues and radiolabeled analogues developed as positron emission tomography (PET) imaging tracers of specific therapeutic small molecule cancer drugs, from published peer-reviewed research reports (abstracts available at www.pubmed.gov). Published literature includes numerous other small molecule radiotracers that have been tested as radiologic biomarkers of important oncotherapeutic targets or as companion diagnostics for radionuclide therapy.

Chemotherapy
C-11 N-[(2′-dimethylamino)ethyl]acridine-4-carboxamide (XR5000) [16] C-11 Temozolomide [17] F-18 5-Fluorouracil [18,19,20,21] F-18 AraG [22] F-18 Clofarabine [23] F-18 Deoxycytidine [24,25,26,27,28] F-18 Paclitaxel [29,30,31,32] F-18 Xeloda [33]
**Anaplastic Lymphoma Kinase (ALK) Inhibitors**
C-11 Lorlatinib [34,35] F-18 Lorlatinib [35]
**Epidermal Growth Factor Receptor (EGFR) Tyrosine Kinase Inhibitors**
C-11 Erlotinib [36,37,38] C-11 Gefitinib [39] C-11 Osimertinib [40] F-18 FEA-Erlotinib [41] F-18 Gefitinib [42] F-18 IRS [43]
**Other Tyrosine Kinase Inhibitors**
C-11 Axitinib [44] C-11 CEP-32496 [45] C-11 Lapatinib [46] C-11 Nintedanib [44] C-11 Sorafenib [47] F-18 Afatinib [38] F-18 Dabrafenib [48] F-18 SKI249380 (Dasatinib Analogue) [9]
**Epichaperome Inhibitor**
I-124 PU-H71 [7,8,49]
**IL2-Receptor/Cytokine**
F-18 IL-2 [50] F-18 FB-IL2v [51]
**Matrix Metalloproteinase Inhibitor**
C-11 (2R)-2-[[4-(6-fluorohex-1-ynyl)phenyl]sulfonylamino]-3-methylbutyric acid methyl ester [52]
**Phosphatidylinositol 3-Kinase (PI3K) Inhibitor**
C-11 Pictilisib (GDC-0941) [53]
**PD-L1**
F-18 BMS-986192 [54]
**Poly(Adenosine Diphosphate Ribose) Polymerase (Parp) Inhibitor**
F-18 Fluorthanatrace [55]

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
