# Peer review of "The Unique Pharmacometrics of Small Molecule Therapeutic Drug Tracer Imaging for Clinical Oncology"

_cancers, 2020, doi:10.3390/cancers12092712_

Round 1

Reviewer 1 Report

This is  a very well written and interesting review on small molecule drug tracer imaging for clinical oncology. 

Author Response

Thank you for your comments.

Reviewer 2 Report

This is an interesting review article for the drug tracer for clinical oncology and concisely summarized. Since imaging is one of the hot topics in clinical oncology, this may provide helpful information. The article is well organized, providing the current status of the topics.

The reviewer has only a few minor concerns as follows;

Table I should provide appropriate references. Check citations. Additional information such as the status of the clinical study and potential as imaging probe is highly appreciated.

Figure 3 does not show (A), (B),… Please confirm the citations.

Author Response

REVIEWER 2 Comments and Suggestions for Authors

This is an interesting review article for the drug tracer for clinical oncology and concisely summarized. Since imaging is one of the hot topics in clinical oncology, this may provide helpful information. The article is well organized, providing the current status of the topics.

The reviewer has only a few minor concerns as follows;

Table I should provide appropriate references. Check citations. Additional information such as the status of the clinical study and potential as imaging probe is highly appreciated.

Authors’ Response: As requested, additional references have been added for each item listed in Table 1.

Figure 3 does not show (A), (B),… Please confirm the citations.

Authors’ Response: The Figure 3 legend has been revised to refer to images by position (instead of A-D).  The citation for the dose escalation therapy trial abstract is correct.

Reviewer 3 Report

This manuscript describes the advantages and potential of small molecule PET imaging. The manuscript is well written and easy to read. I would like to suggest the following to improve the manuscript:

  1. Please also describe in more extent the limitations of small molecule PET imaging. I would favor the opinion that PET imaging and classical PK/PD studies are not competitors, but can add improtant information to each other and that data should be combined during early phase drug development.
  2. Although it is already described in the text, in its current form it is difficult for the reader to extract how a new tracer should be developed. I would like to see a separate paragraph or figure where the authors guide the reader how a new tracer should be developed to aid drug development. I hope this is possible. 

Author Response

REVIEWER 3 Comments and Suggestions for Authors

This manuscript describes the advantages and potential of small molecule PET imaging. The manuscript is well written and easy to read. I would like to suggest the following to improve the manuscript:

  1. Please also describe in more extent the limitations of small molecule PET imaging. I would favor the opinion that PET imaging and classical PK/PD studies are not competitors, but can add important information to each other and that data should be combined during early phase drug development.

Authors’ Response: The Authors agree with the Reviewer’s comment. Classical plasma and blood-based PK assays and blood or other tissue-based PD studies offer vital information that PET-based PK and biodistribution imaging cannot, including learning drug metabolism, oral bioavailability, drug and food drug interactions, etc. This raises the question, ‘Then why do we need PET imaging or what can it offer?’, which is the focus of the manuscript. Because of the prescribed limitation of word count, we removed much of the review of what classical PK/PD offers.

Clinical development of PET imaging biomarkers is hindered by its considerable economic costs, including its preclinical and early clinical phases (eg, radiology and cyclotron infrastructure), multidisciplinary expertise (radiochemistry, nuclear medicine physic radiotracer dose required multi intrinsic technical and logistical demands of using investigational radiopharmaceuticals

  1. Although it is already described in the text, in its current form it is difficult for the reader to extract how a new tracer should be developed. I would like to see a separate paragraph or figure where the authors guide the reader how a new tracer should be developed to aid drug development. I hope this is possible. 

Authors’ Response: As requested, we have added a new section describing the development of a new drug tracer. This required four paragraphs.